# Enhancer RNA Expression in Response to Glucocorticoid Treatment in Murine Macrophages

**DOI:** 10.3390/cells11010028

**Published:** 2021-12-23

**Authors:** Franziska Greulich, Kirsten Adele Bielefeld, Ronny Scheundel, Aikaterini Mechtidou, Benjamin Strickland, Nina Henriette Uhlenhaut

**Affiliations:** 1Metabolic Programming, TUM School of Life Sciences, ZIEL Institute for Food & Health, Gregor-Mendel-Strasse 2, 85354 Freising, Germany; franziska.greulich@tum.de (F.G.); r-scheundel@tum.de (R.S.); benjamin.strickland@tum.de (B.S.); 2Helmholtz Diabetes Center (IDO, IDC, IDE), Helmholtz Center Munich HMGU, Ingolstaedter Landstr. 1, 85764 Neuherberg, Germany; kirsten.bielefeld@alum.utoronto.ca (K.A.B.); aikaterini.mechtidou@helmholtz-muenchen.de (A.M.)

**Keywords:** glucocorticoid receptor, inflammation, enhancer RNA, macrophages, tissue specificity, transcription

## Abstract

Glucocorticoids are potent anti-inflammatory drugs; however, their molecular mode of action remains complex and elusive. They bind to the glucocorticoid receptor (GR), a nuclear receptor that controls gene expression in almost all tissues in a cell type-specific manner. While GR’s transcriptional targets mediate beneficial reactions in immune cells, they also harbor the potential of adverse metabolic effects in other cell types such as hepatocytes. Here, we have profiled nascent transcription upon glucocorticoid stimulation in LPS-activated primary murine macrophages using 4sU-seq. We compared our results to publicly available nascent transcriptomics data from murine liver and bioinformatically identified non-coding RNAs transcribed from intergenic GR binding sites in a tissue-specific fashion. These tissue-specific enhancer RNAs (eRNAs) correlate with target gene expression, reflecting cell type-specific glucocorticoid responses. We further associate GR-mediated eRNA expression with changes in H3K27 acetylation and BRD4 recruitment in inflammatory macrophages upon glucocorticoid treatment. In summary, we propose a common mechanism by which GR-bound enhancers regulate target gene expression by changes in histone acetylation, BRD4 recruitment and eRNA expression. We argue that local eRNAs are potential therapeutic targets downstream of GR signaling which may modulate glucocorticoid response in a cell type-specific way.

## 1. Introduction

Endogenous glucocorticoids (GCs) such as cortisol (humans) or corticosterone (rodents) are essential hormones released in a diurnal fashion and controlled by the hypothalamus-pituitary-adrenal axis. In humans, circulating glucocorticoid levels peak in the morning, coincident with the anticipation of food intake, and decline during the course of the day [1,2,3]. GCs regulate metabolism, behavior, cellular differentiation, circadian clocks and immune processes [3,4,5,6]. Synthetic glucocorticoids such as dexamethasone or prednisolone, on the other hand, are widely prescribed drugs for the treatment of inflammatory and autoimmune diseases [7]. Most recently, GCs have become the gold standard for the treatment of severe COVID-19 [8]. However, their long-term application is accompanied by severe side effects which ultimately limit their clinical usefulness [9,10].

GCs bind the glucocorticoid receptor (GR), encoded by the *Nr3c1* gene. The GR is an essential and ubiquitously expressed nuclear hormone receptor [11]. Upon ligand binding, GR translocates to the nucleus, scans the chromatin landscape for GR binding sites, and activates or represses target genes by binding to promoters or enhancers [4,12,13,14,15,16]. Transcriptional repression by GR is mostly associated with its desired anti-inflammatory and immunomodulatory functions [4,16]. On the other hand, gene activation is thought to largely mediate GR’s metabolic functions, which may contribute to the undesired side effects [3]. GR has been shown to interact with various co-regulators such as NCoAs or SRCs, P300/CBP and the Mediator complex to activate its target genes [3,12,17,18,19,20,21,22]. Concurrently, histone acetylation is a chromatin mark deposited by histone acetyl transferases such as P300/CBP, which is widely associated with active gene expression [23,24].

In contrast, gene repression is less well understood and may involve the interplay of several mechanisms; among them are competitive, cooperative and DNA-independent scenarios of negative regulation [18,25,26]. For instance, we recently found that gene repression, indeed, depends on DNA binding by GR, whereas GR’s interaction with chromatin at repressive sites may not [12]. GR has been shown to recruit corepressors such as GRIP-1 (SRC2) or NCoR/SMRT and, subsequently, HDACs to reduce histone acetylation marks at repressive binding sites [13,18,27].

Besides the diversity in transcriptional mechanisms, GR target genes are regulated in a highly cell type-specific manner due to the requirement for pre-determined, accessible cis-regulatory sites. Such available sites represent open chromatin, established by pioneer factors, nucleosome remodelers and P300/CBP during lineage specification, which enable GR binding [28,29,30,31,32,33]. Conceivably, these tissue-specific GR target gene programs confer the beneficial anti-inflammatory effects of glucocorticoid therapy in macrophages, for example, as opposed to the adverse metabolic side effects in, e.g., hepatocytes [29,33]. Therefore, a mechanistic understanding of tissue-specific mediators and modulators of GR signaling is essential to optimize GC therapy in the future.

Here, we used 4-thioluridine (4sU) metabolic labelling to measure the direct, nascent transcriptional output upon GC stimulation. We identified enhancer RNAs (eRNAs), nascent and unstable transcripts produced at intergenic GR binding sites, as potential targets to fine tune cellular GC responses. We show that synthetic glucocorticoids such as dexamethasone (Dex) are able to induce and repress eRNA expression in murine bone marrow-derived macrophages during inflammatory activation by lipopolysaccharide (LPS). GR-mediated eRNA expression highly correlates with the expression of GR target genes and tissue-specific GC function. We further associate eRNA expression in LPS-activated macrophages with histone 3 lysine 27 (H3K27) acetylation and BRD4 binding at GR-occupied intergenic enhancers.

## 2. Material and Methods

### 2.1. Animals

C57BL6/J mice were housed in a controlled SPF facility with a 12 h light/dark cycle at 23 °C with constant humidity, and fed were ad libitum. Mouse experiments were performed in accordance with HMGU guidelines for the care and use of animals, overseen by the district government of Upper Bavaria. We exclusively used male mice aged 5–8 weeks for the isolation of bone marrow-derived macrophages.

### 2.2. Isolation of Bone Marrow-Derived Macrophages

Primary bone marrow-derived macrophages were isolated from 5–8 week-old male mice (C57BL6/J). The bone marrow was flushed from humerus, femur and tibia with RPMI and erythrocytes lysed in AKC lysis buffer (1 M NH_4_Cl, 1 M KHCO_3_, 0.5 M EDTA). Cells were further purified by Ficoll gradient (Ficoll-Paque Premium; GE Healthcare, Chicaogo, IL, USA, Cat. No.:17-5442-02) and differentiated within 7 days in DMEM (4.5 g/L glucose, Sigma-Aldrich, St. Louis, MO, USA) supplemented with 20% FBS (Sigma-Aldrich), 1% penicillin/streptomycin (Sigma-Aldrich) and 30% L929 cell supernatant, as previously described [4,19]. Differentiated macrophages were cultured in serum-free macrophage medium (ThermoFisher Scientific, Waltham, MA, USA, Cat. No.:12065074).

### 2.3. Treatment and 4sU Labelling

Twenty million primary bone marrow-derived macrophages were plated on 15-cm cell culture dishes in serum-free macrophage medium. Cells were pre-treated with either 1 μM dexamethasone (Dex, Sigma-Aldrich, Cat. No.: D4902) or vehicle (Ethanol) for 30 min, followed by 2 h of 100 ng/mL LPS treatment (LPS E. COLI O111:B4, Sigma-Aldrich, Cat. No.: LPS25). After 1 h of LPS treatment, 200 μM of 4-thiouridine (4sU, Carbosynth Ltd., UK, Cat. No.: 13957-31-8) was added to label nascent transcripts. Note, the 4-thiouridine concentration and incubation time were optimized concerning cell viability and incorporation efficiency as determined by dot plot assays (data not shown). All treatments were performed in dark humidified incubators at 37 °C and 5% CO_2_. After the treatment, the cell culture medium was removed, and cells were washed in D-PBS and collected in 10 mL Trizol (Invitrogen, Waltham, MA, USA) per 15-cm dish.

### 2.4. RNA Isolation and 4sU-Seq

4sU labelling and purification of nascent transcripts was performed according to the protocol published by Raedle at al. [34]. In detail, RNA from Trizol-lysed and 4sU-labelled macrophages was isolated by chloroform-phenol extraction according to standard protocols. For Thiol-specific biotinylation of the nascent (4sU-incorporated) RNA, 80 μg of total RNA were incubated with 1 mg/mL EZ-Link Biotin-HPDP (solubilized in DMF, Pierce, Cat. No.: 21341) in biotinylation buffer (10 mM Tris pH = 7.4, 1 mM EDTA) for 1.5 h. Subsequently, the total RNA was purified by chloroform-phenol extraction and precipitated with isopropanol in high salt conditions. The total RNA was diluted to 1 μg/mL and denatured (10 min 65 °C). Streptavidin pulldown using streptavidin-coated magnetic beads (Miltenyi Biotec, Bergisch Gladbach, Germany, Cat. No.: 130-074-101) and μMACS columns (Miltenyi Biotec, Cat. No.: 130-074-101) was performed to capture the fraction of nascent (biotinylated) RNAs. The nascent RNA was purified using Agencourt RNA clean XP beads (Beckman Coulter, Brea, California, USA, Cat. No.: A63987). The concentration of the nascent RNA was quantified using Qubit2.0 (ThermoFisher Scientific), and the RNA quality was judged by Bioanalyzer (Bioanalyzer2100, Agilent Technologies, Santa Clara, CA, USA) using the RNA 6000 Nano Reagents (Agilent Technologies, Cat. No.: 5067-1511). Depletion of rRNAs (Ribo-Zero Gold rRNA Removal Kit (Illumina, San Diego, CA, USA)) and library preparation was performed starting with 400 ng of nascent RNA using the TruSeq Stranded Total RNA Library Prep Kit (Illumina, total RNA) or the TrueSeq Stranded mRNA Library Prep Kit (Illumina, PolyA-RNA). Amplified cDNA libraries were further purified using Agencourt RNAClean XP Beads and quality control of biotinylated RNA and cDNA libraries were performed using Agilent Bioanalyzer 2100 with the RNA 6000 Nano Reagents (Agilent Technologies, Cat. No.: 5067-1511) or High Sensitivity DNA Reagents (Agilent Technologies, Cat. No.: 5067-4626). Paired-end sequencing was performed on the HiSeq2500 (Illumina) with 100 bp read length.

### 2.5. RNA Isolation and Quantitative Real-Time (qRT-) PCR

Total RNA from 200,000 cells was isolated using the SV Total RNA Isolation kit (Promega, Madison, WI, USA, Cat. No. Z3105) and 500 ng RNA were reverse transcribed with the Reverse Transcription System from Promega (Cat. No.: A3500) according to the manufacturer’s manual. Importantly, DNA contaminations were removed via DNaseI treatment. Quantitative PCRs with primers against murine genes or genomic loci *Tsc22d3/Gilz* (*Tsc22d3*_F: ACCACCTGATGTACGCTGTG; *Tsc22d3*_R: TCTGCTCCTTTAGGACCTCCA, T_A_ = 60.0 °C, 51 bp product), *Tsc22d3/Gilz* eRNAs 1 (*Tsc22d3*_eRNA1_F: TTGAGCAGCAGACACGTTCA; *Tsc22d3*_eRNA1_R: GAATTCCTCCGACGGGAACT, T_A_ = 60.2 °C, 189 bp product), 2 (*Tsc22d3*_eRNA2_F: AACACAGGGTGGACATTGGG; *Tsc22d3*_eRNA2_R: ATGGTGGTGGAAAGCAAGGT, T_A_ = 60.2 °C, 73 bp product) and 3 (*Tsc22d3*_eRNA3_F: GGAGTTGGG TTCTGCCTGAG, *Tsc22d3*_eRNA3_R: CCTGACCCTGTCTTTCCAGTG, T_A_ = 60.3 °C, 52 bp product), *Ccr3* (Ccr3_F: GAAAACTTGCAAAACCTGAGAAGC; Ccr3_R: TGCCATTCTACTTGTCTCTGGTG, T_A_ = 59.7 °C, 174 bp product), *Ccr3* eRNA (Ccr3_eRNA_F: TGTAGAGCAGAGGGCTGGAT, Ccr3_eRNA_R: CCAGGTTACAGTGCCCCATT, T_A_ = 60.0 °C, 98 bp product) and the housekeeping gene *18S* (18S_F: AAACGGCTACCACATCCAAG, 18S_R: CCTCCAATGGATCCTCGTTA, T_A_ = 58.2 °C, 155 bp product) were performed using the GoTaq qPCR master mix (Promega, Cat. No.: A6002) and CFX96 Touch Deep Well Real-Time PCR System (Bio-Rad Laboratories, Hercules, CA, USA). Primers were designed to have annealing temperatures (T_A_) between 58 and 61 °C (Eurofins). Experiments were performed with three biological and three technical replicates.

### 2.6. NGS Data Analysis

#### 2.6.1. Data Sources

ChIP-seq against GR, H3K4me1/me2/me3, H3K27ac, SETD1A, CXXC1, ATAC-seq and RNA-seq in bone marrow-derived macrophages was previously published by our lab and is available at NCBI’s GEO [35,36]. Similarly, our lab previously released ChIP-seq data for GR (GSM2911260 and GSM2911261 [3]) and RNA-seq data from murine liver tissue (GSE108690 [3]).

We further used publically available ChIP-seq data for RNAP2 and BRD4 in murine bone marrow-derived macrophages (GSE109131 [16]) and DNaseI HyperSensitivity (DHS) (GSM1479704 [37]) as well as GRO-seq data for murine livers (GSM1437739 [38]). An overview of the data including individual accession numbers and exact treatment conditions appears in Appendix A.

#### 2.6.2. RNA Sequencing (RNA-Seq)

For RNA-seq, gene-level quantification was performed with *Salmon* version 1.4.0 (RRID:SCR_017036 [39]). Settings were -libType A, -gcBias and -biasSpeedSamp 5 using the mm10 (M25, GRCm38) reference transcriptome provided by Genecode [40]. Gene count normalization and differential expression analysis were performed with *DESeq2* version 1.32.0 (RRID:SCR_015687 [41]) after import of gene-level estimates with *tximport* version 1.20.0 (RRID:SCR_016752 [42]) in *R* version 4.1.0 (RRID:SCR_001905 [43]).

For gene annotation, Ensembl gene identifiers were mapped to MGI symbols using the Bioconductor package *biomaRt* version 2.48.2 (RRID:SCR_002987 [44]) and genome information was provided by *Ensembl* (GRCm38.p6 [45]). Genes with at least 1 read count, fold change of 1.5 and Benjamini-Hochberg-adjusted *p* value < 0.05 were called significantly changed.

#### 2.6.3. sU-Seq

NGS data quality was assessed with *FastQC* (RRID:SCR_014583, http://www.bioinformatics.babraham.ac.uk/projects/fastqc/, accessed on 10 December 2021). TrueSeq adapter sequences and low-quality reads were trimmed using *Trimmomatic* version 0.39 (RRID:SCR_011848 [46]). Paired-end reads from 4sU-labelling (macrophages) or global run on sequencing (GRO-seq, liver GSM1437739 [38]) experiments were mapped to the murine genome build GRCm38.p6 (*Ensembl*, [45]) using *HISAT2* version 2.2.1 (RRID:SCR_015530 [47]). Transcripts were de-novo assembled using *StringTie* version 2.1.7b (RRID:SCR_016323 [48]) and transcript annotations provided by *Ensembl* (GRCm38.p6 [45]) with the following settings: --rf -f 0.1 -c 2 -A –B –m 50. Transcripts assembled from all three replicates of LPS and LPS plus Dex-treated macrophages were merged using *StringTie* with the following parameters, -m 50 -T 0.01 -F 0.01, and were compared to annotated transcripts (GRCm38.p6) with *gffcompare* [49]. Genome browser tracks were scaled according to the DESeq normalization factor and mean BigWig files per treatment generated with *deepTools* version 3.5.0 (RRID:SCR -016366 [50]). Genome browser tracks were visualized with the *UCSC Genome Browser* (RRID:SCR_005780 [51]). DESeq normalization factors were computed from all assembled transcripts after transcript quantification with *featureCounts* version 2.0.1 (RRID:SCR_012919 [52]) and subsequent differential expression analysis with *DESeq2* version 1.32.0 (RRID:SCR_015687 [41]).

For differential expression analysis at GR enhancer, fragments overlapping 1 kb around all reproducible GR binding sites (GBSs) were counted with *featureCounts* version 2.0.1 (RRID:SCR_012919 [52]) with the following parameters: -*p* -Q 20 -B -d 50 -D 1000 -s 0 -C -T 14. Counts were supplied as input for *DESeq2* version 1.32.0 (RRID:SCR_015687 [41]). The size factors computed by *DESeq2* were replaced by the size factors estimated from the differential RNA-seq analysis performed for all assembled transcripts (see above). The results were filtered for eGBSs identified as described below (“Identification and characterization of enhancer RNAs“), and the differential enrichment was visualized as volcano plot using *ggplot2* version 3.3.5 (RRID:SCR_014601 [53]).

#### 2.6.4. Defining the Reproducible GR Cistrome

Reproducible GR binding sites were identified after mapping two replicates of GR ChIP-seq data (macrophages: GSM5654976 and GSM5654977 [36], liver: GSM2911260 and GSM2911261 [3]) to the murine genome build GRC38.p6 (mm10, [45]) using *BWA-MEM* version 0.7.13 (RRID:SCR_010910 [54]). PCR duplicates were labelled using *Picard tools* version 2.01.1 (RRID:SCR_006525, https://broadinstitute.github.io/picard/, accessed on 10 December 2021). Unmapped/unpaired reads and PCR duplicates were removed using *Samtools* version 1.11 (RRID:SCR 002,105 [55]). Peaks were called over matched input controls (macrophages: GSM4040461 [19], liver GSM2911286 [3]) using *MACS* version 3.0.0a5 in BAMPE mode with a FDR threshold of 0.1 (RRID:SCR_013291 [56]). Blacklisted regions (http://mitra.stanford.edu/kundaje/akundaje/release/blacklists/mm10-mouse/mm10.blacklist.bed.gz, accessed on 10 December 2021) were removed from the peak set [57].

GR peaks with a genomic overlap of at least 1 bp between the two replicates were called reproducible and used for subsequent analyses. Reproducible GR peaks were annotated to the closest expressed gene with the *ChIPpeakAnno* package (version 3.24.1, RRID:SCR_012828 [58]) in *R* (version 4.1.0 [43]) using the *UCSC*-provided known gene annotation for the genome build mm10 [59]. GR binding sites were termed intergenic when not overlapping any gene feature (exon, promoter, intron, UTR, etc.) and positioned more than 1 kb from any transcriptional start site.

#### 2.6.5. Identification and Characterization of Enhancer RNAs

For the transcript-based approach (see Figure 1A), nascent transcript libraries from the 4sU-labelling experiment were submitted for total RNA-seq and PolyA-RNA-seq. Paired-end data for both library strategies were processed the same way and transcripts were assembled with *StringTie* (version 2.1.7b, RRID:SCR_016323 [48]) as described above. Reads for each transcript were counted using *featureCounts* version 2.0.1 (RRID:SCR_012919 [52]) with the following parameters: -*p* -Q 20 -B -d 50 -D 1000 -s 0 -C -T 14. Transcripts per million (TPMs) were calculated as follows: (counts per transcript *1000)/(library size * transcript length). We called transcripts ‘expressed’ which passed the 5th percentile of the mean expression across all samples (*n* = 3 for LPS and LPS + Dex treatments each). Transcripts were annotated by proximity to the closest expressed gene using the *ChIPpeakAnno* package (version 3.24.1, RRID:SCR_012828 [58]) in *R* with *UCSC*-provided known gene annotations for the genome build mm10 [59]. Transcripts not overlapping any gene feature and more than 1 kb away from any TSS were termed intergenic. Transcripts that were identified in the total and PolyA-enriched libraries were designated ‘polyadenylated’.

For the GR binding site (GBS)-centered approach, we counted those reads overlapping reproducible GR peaks within the range of 4 nucleosomes (588 bp) using *BEDtools* version 2.27.1 (RRID:SCR_006646 [60]). TPMs were calculated as above using 588 bp as transcript length. Intergenic GBS were identified as above and GBS were called eRNA-producing (eGBS) when the expression level passed the 5th percentile of the mean expression level for all reproducible GBSs across conditions (*n* = 3 for LPS and LPS + Dex each).

Overlaps between eGBSs identified with the transcript-based and GBS-based approach were visualized using the *VennDiagram* package version 1.6.20 in *R* (RRID:SCR_002414). Overlaps were defined at 1 bp resolution. The bar plot reflecting the intersection sizes of eGBSs with DNA accessibility, H3K27 acetylation and H3K4 mono-methylation was visualized with the *upsetR* package version 1.4.0 [61] in *R*. Only peaks reproducibly called in at least two replicates were used for the analysis. Two peak sets were intersecting when they overlapped for 10 bp.

#### 2.6.6. DNA Motif Enrichment

For all DNA motif analyses, peaks were trimmed to 100 bp around the peak center and repeats were masked using the *RepeatMasker* tool integrated into *UCSC* (RRID:SCR_012954 [62], RRID:SCR_005780 [51]). Quantification of DNA motif occurrence was performed with *FIMO* version 3.5.0 [63] using the *JASPAR* [64] reference motifs MA0113.1/2/3 (NR3C1), MA0080.3 (SPI1), MA0099.1 and MA1130.1 (AP1), MA0107.1 (RELA), MA1107.2 (KLF9), POL012.1 (TATA), POL005.1 (DPE), POL006.1 (BREu), POL007.1 (BREd), POL008.1 (DCE_I), POL009.1 (DCE_II), POL010.1 (DCE_III), POL011.1 (XCPE1), POL002.1 (INR) and POL001.1 (MTE). Bar plots were generated using *R* version 4.1.0 [43] and the *ggplot2* package version 3.3.5 (RRID:SCR_014601 [53]).

Motif enrichment without a background set was performed with *MEME* suite version 3.5.0 (RRID:SCR_001783 [65]) using the following parameters: -dna -meme-mod zoops -minw 5 -maxw 25 -meme-nmotifs 20 -meme-*p* 10, and the motif databases *JASPAR* (2018 version, RRID:SCR—003,030 [64]), *Uniprobe* (RRID:SCR_005803 [66]) and *SwissRegulon* (RRID:SCR 005,333 [67]).

Differential motif enrichment was performed using *XSTREME* version 5.4.1 [68] with the following settings: --dna --evt 0.05 --minw 5 --maxw 20 --align center --meme-mod zoops, and the motif databases JASPAR (2018 version, RRID:SCR_003030 [64]) and Uniprobe (RRID:SCR_005803 [66]).

#### 2.6.7. Gene Ontology (GO) Enrichment for Biological Processes

GO enrichment analysis was performed with the *clusterProfiler* package version 3.18 (RRID:SCR 016,884 [69]). GO terms with 60% redundance in gene content were removed and the top 10 non-redundant biological processes with a *q*-value < 0.001 for each subset were displayed as bar plots with *ggplot2* version 3.3.5 (RRID:SCR_014601 [53]).

#### 2.6.8. Analysis of ChIP Sequencing (ChIP-Seq) Data and Quantification

Publicly available ChIP-seq data were processed as mentioned above (“Defining the reproducible GR cistrome”). ATAC-seq and DNaseI hypersensitivity (DHS) data were analyzed as previously described [36].

For the quantification of read density at eGBSs, reads were counted at 1 kb around the reproducible GR binding sites with *BEDtools* version 2.27.1 (RRID:SCR_006646 [60]). Unscaled, de-duplicated BAM files were used as input. Differential binding analysis was performed with *DESeq2* version 1.32.0 (RRID:SCR_015687 [41]). We replaced the size factors used in the normalization step of *DESeq2* by the reciprocal of the reads-in-peaks ratio (RiP ratio, see Appendix A) of each ChIP-seq dataset over its peak union (peaks detected across all replicates and conditions) scaled by the smallest dataset, whose RiP ratio was set to 1. This approach allowed us to normalize to library size as well as signal-to-noise ratio. The same RiP ratios were used to scale individual coverage tracks used to compute mean genome browser tracks and heatmaps with *deepTools* version 3.5.0 (RRID:SCR_016366 [50]). The correlation plot of chromatin features and eRNA expression at differentially expressed eGBSs was generated with the *corrPlot* package version 0.90 [70] in *R* version 4.1.0. Violin plots were visualized with *ggplot2* version 3.3.5 (RRID:SCR_014601 [53]). The significance of the eRNA expression contribution towards each chromatin feature was tested with the Kolmogorov–Smirnov test and, if significant, by subsequent pair-wise Wilcoxon–Mann–Whitney test, using the unchanged subset as reference.

## 3. Results

### 3.1. Intergenic Glucocorticoid Receptor Binding Coincides with eRNA Synthesis in Macrophages

In order to study the transcriptional events mediating GR’s anti-inflammatory effects, we used primary murine bone marrow-derived macrophages activated with the TLR4 ligand lipopolysaccharide and treated with the GR ligand Dexamethasone (LPS versus LPS + Dex). With the aim of identifying those GR binding sites (GBS) exhibiting detectable nascent transcription, we used two strategies (Figure 1A): first, we took a transcript-based approach and labelled nascent RNA species with 4-thiouridine (4sU) for 1 h during the LPS or LPS + Dex treatments. Nascent transcripts were isolated by subsequent biotinylation and streptavidin pulldown and submitted for total RNA sequencing.

With this approach, we identified 267,394 nascent transcripts by mapping to the reference genome (mm10, GRC38.p6) followed by transcript assembly using StringTie (Figure 1A [48]). After the removal of lowly expressed labelled transcripts (TPMs > fifth percentile of non-zero mean expression counts), 189,692 nascent transcripts remained. As expected, most of the 4sU-labelled transcripts mapped to gene features such as promoters, introns, exons or UTRs. The remaining 5.5% (10,451) of nascent transcripts did not map to or overlap with any known gene (Genecode version M25 [40]) and were more than 1 kb away from any annotated transcriptional start site (TSS). In contrast to the total number of 4sU-labelled transcripts, of which 57.6% were also identified in a PolyA-selective RNA-seq experiment, only 0.2% of the intergenic nascent RNAs (iRNAs) could be detected in PolyA-enriched RNA-seq libraries (Appendix A). Among these iRNAs, we selected those aligning to known intergenic GR binding sites that we had mapped by GR ChIP-seq in primary macrophages (iGBS, [4,19,36]). Thus, 860 nascent transcripts were identified which occurred at 630 intergenic GR-bound loci (Figure 1A,B). These iRNAs are termed GR enhancer RNAs (eRNAs) henceforth.

Interestingly, the length distribution of GR eRNAs were skewed towards longer transcripts when compared to the length distribution of iRNAs not mapping to a GBS (Appendix A). Of all iRNAs, ~61% were composed of one exon, ~31% were predicted to have two exons and ~8% comprised more than two exons (Appendix A), indicating that a minor subset of those nascent transcripts might rather represent unannotated long non-coding RNAs. The majority of GR eRNAs identified by this approach were predicted to be composed of two exons, followed by a number of mono-exonic nascent transcripts (Appendix A). The higher amount of predicted two-exonic transcripts among the GR eRNAs might explain the shift towards longer RNAs shown in Appendix A.

In a second approach, we took a GR-centric point of view and counted all the reads from our nascent transcriptomes mapping within 1 kb around the 8565 reproducible GR ChIP-seq peaks in LPS plus Dex-treated bone marrow-derived macrophages (Figure 1A). We removed GBSs with zero reads counts as well as nascent transcripts with expression below the fifth percentile of the non-zero mean counts across all GBSs and conditions. The remaining 7117 GBS (~84%) were filtered for their intergenic location (+/−1 kb from any TSS). Twenty-three point seven percent of all GBSs with detectable nascent RNA expression were intergenic, whereas the majority of the GBS were located in introns (Figure 1A, right). With the GR-based approach, we were able to identify almost all of the GR eRNAs from the transcript-based approach as well as 1070 additional GBSs displaying nascent transcription (Figure 1C). The GR-based approach was more sensitive than the transcript-based approach, as it did not rely on transcript assembly and was less stringent in terms of expression filtering (due to the lower fraction of highly expressed mRNAs, see Figure 1A). In summary, we identified nascent transcripts at 69.2% of intergenic GR binding sites (Figure 1B right).

Next, we asked what distinguished eRNA-producing from ‘nonproductive’ GBSs by analyzing the DNA motifs 100 bp around the GR ChIP peak center. An enrichment analysis using MEME-ChIP [65] identified SPI1/PU.1 and AP1 motifs in the GBS subset expressing nascent transcripts. Similarly, SPI1/PU.1, AP1, zinc finger and GRE (Glucocorticoid Response Element) motifs were detected in the non-iRNA-producing subset of GR binding sites (Appendix A). We analyzed the occurrence of each of these motifs as well as the proportion of known promoter elements such as TATA boxes, downstream promoter elements (DPEs), upstream (u) and downstream (d) transcription factor II B response elements (BREs), motif ten elements (MTE), X core promoter element 1s (XCPE1) and initiator motifs (INR) among each GBS subset [71]. Around 30% of the GBSs with and without eRNAs contained the GRE motif, especially the motif sequences 2 and 3 (MA0113.2/3 [64], Figure 1D), the latter being slightly underrepresented in the group of GBSs with eRNAs, consistent with the MEME analysis (Appendix A). The AP1 motif was identified in 28.8% of the GBS expressing eRNAs, but only in 17.9% of the cases without eRNAs, which may represent the most prominent difference between the two subsets (Figure 1D). However, these differences were not significant in differential motif enrichment analyses (data not shown). Among the promoter elements, we observed a tendency towards an increased abundance in the GBS subset producing eRNAs. However, all but the INR and XCPE1 motifs were identified in less than 10% of sites, confirming that most of the GR binding events do not map to core promoter elements (Figure 1D).

In addition to the DNA sequence itself, the chromatin environment shapes the transcriptional landscape. Histone modifications, especially, were shown to serve as marks of active transcription by providing docking platforms for chromatin modifiers, structural proteins, the transcriptional machinery and a myriad of co-regulators [72,73,74]. Therefore, we analyzed the chromatin environment around the GBSs with eRNA expression (eGBSs) by integrating publicly available data. Sixty-five point eight percent of the eGBSs were concomitantly marked by high DNA accessibility, as determined by ATAC-seq, histone 3 lysine 4 mono-methylation (H3K4me1) and histone 3 lysine 27 acetylation (H3K27ac) as profiled by ChIP-seq (Figure 1E) [19,36]. These features likely reflect the active enhancer status of these binding sites. Conversely, 28.7% of the eGBSs did not display any H3K27ac marks, but still showed enrichment of H3K4me1 and DNA accessibility, marking those cis-regulatory elements as “poised” (Figure 1E) [23].

Example genome browser tracks for the activated GR target gene *Tsc22d3* (*Gilz*) and the repressed targets *Ccr3* and *Ccr2* show that GR binding sites with eRNA expression (4sU labelled reads) overlap with regions of DNA accessibility (ATAC-seq), H3K4me1 and H3K27ac (Figure 1F).

In summary, our 4sU labeling identified intergenic GR binding sites showing quantifiable eRNA expression in activated macrophages. These GR occupied sites producing eRNAs enriched for AP1 as well as GRE motifs and were marked by classical enhancer features such as open chromatin, H3K4me1 and, partially, H3K27ac.

### 3.2. Enhancer RNA Synthesis at GR Binding Sites Is a Tissue-Specific Feature of the Glucocorticoid Response

GR binding and target gene regulation are distinctly cell type specific and lineage-dependent, and glucocorticoid responses have been shown to be pre-determined by chromatin accessibility [28,29,30,31,32,75]. To address the question of whether glucocorticoid-induced eRNA expression is also tissue-specific, we compared our macrophage data sets to published data from mouse livers collected at the peak of corticosterone release (ZT13, GSM1437739 [38]).

First, we confirmed the tissue-specific response to glucocorticoids by differential mRNA expression analysis comparing LPS- versus LPS + Dex-treated murine macrophages [19] and livers harvested at the peak versus the trough of glucocorticoid secretion (ZT12 and ZT0) [3]. Indeed, we found 72.7% of hepatic genes and 95.6% of myeloid genes to be regulated by glucocorticoids in a tissue-specific manner (Appendix A), which may reflect their biological function in the respective cell types [4,29,33,76]. Macrophage-specific genes are involved in inflammatory processes, whereas liver-specific differential genes enrich for metabolic pathways (Appendix A, Appendix A). Among the few genes shared across both tissues, we identified ‘circadian rhythms’ as the common, ubiquitous biological pathway (Appendix A, Appendix A).

Next, we processed publicly available GRO-seq data for murine livers harvested at the peak of corticosterone release (ZT13, GSM1437739 [38]) in the same way as our macrophage 4sU-Seq data (see methods). eRNA-expressing, intergenic GBSs were identified by quantifying the GRO-seq signal at 1 kb around liver GR ChIP peak centers (ZT12 [3]). GBSs were defined as eRNA-producing GBSs (eGBSs) if the GRO-seq counts passed the fifth percentile of the mean expression at all sites, which left 1548 eGBSs (Appendix A).

Comparing macrophage and liver eGBSs, we only found 38 common, productive cis-regulatory sites (Appendix A). However, when we investigated the genes closest to these eGBS, we found 197 genes in common between liver and macrophages, suggesting that a small subset of tissue-specific eGBSs might converge to regulate similar genes (Figure 2A). Nevertheless, over 97% of the eGBS and 80.6% or 82.9% of the genes associated with eGBS in macrophages and livers, respectively, were expressed in only one of the two tissues (Figure 2A). The GBS-associated genes, again, reflected the cell type-specific functions of glucocorticoids [29,33,76]. Genes in proximity to macrophage-specific eGBSs related to inflammatory pathways such as “regulation of T cell activation,” “positive regulation of cytokine production” and “TNF production”. Liver-selective eGBSs, on the other hand, were associated with pathways such as “fatty acid metabolic process,” “cellular carbohydrate metabolic process” and “regulation of lipid biosynthetic process” (Figure 2B, Appendix A). Interestingly, we did not find an enrichment for circadian genes among those with shared association to eGBSs in macrophages and livers. Shared eGBSs, rather, enriched for “fat cell differentiation,” “myeloid cell differentiation” and “response to LPS” (Figure 2B).

As previously described, GR binding, as well as enhancer accessibility, is highly lineage-dependent [28,31,32]. Therefore, we compared available ATAC-seq and GR ChIP-seq data from activated macrophages (LPS + Dex) (GSE186511 and GSE1865112 [36]) with GR ChIP-seq (GSM3703244 [3]) and DNaseI HyperSensitivity (DHS) data from liver tissue (GSM1479704 [37]) at the peak of glucocorticoid secretion (ZT12/14). The comparison of the GR cistromes from macrophages and livers confirmed that over 90% of the sites were bound selectively (Appendix A). Similarly, accessible regions as profiled by ATAC-seq in macrophages treated with LPS plus Dex or by DNase I-hypersensitivity sequencing in livers at ZT14 showed 65–69% tissue specificity (Appendix A), which presumably might provide the basis for selective GR occupancy [28,31,32]. For macrophage- versus liver-specific eRNA producing sites, we detected distinct GR binding as well as open chromatin signatures (Figure 2C). However, eRNA-transcribing loci found in both macrophages and livers also shared GR binding and accessibility (Figure 2C).

Differential motif enrichment of the myeloid-specific eGBS subset over the hepatic eGBS subset identified the lineage factor SPI1/Pu.1 [77] and the inflammatory transcription factors AP-1 and NF-κB as enriched in macrophages (Figure 2D), validating previous analyses [78,79]. Vice versa, the hepatocyte lineage factors HNF4 and FoxA co-occurred at liver-specific eGBSs (Figure 2D) [80,81]. Among the shared GBSs, we identified motifs for AP-1 as well as the C2H2 zinc finger containing krüppel-like factor (KLF) family as commonly enriched over either the macrophage-specific or liver-specific sequencing reads (Figure 2D). One example of one macrophage-specific, one shared and one liver-specific eGBS is shown in Figure 2E. Note that common eGBSs are located within DNA accessible regions in both tissues, whereas tissue-specific eGBSs mostly do not (Figure 2E).

In summary, we found that tissue-specific GR binding as a consequence of lineage-dependent accessibility of the binding site correlates with tissue-specific eRNA transcription and glucocorticoid responses.

### 3.3. Differential Expression of eRNAs upon Glucocorticoid Treatment in Macrophages

Since we had identified a set of macrophage-specific GR-bound loci displaying noncoding transcription, we next examined if GR binding indeed changed the expression of eRNAs from these sites. Therefore, we analyzed our 4sU-seq profiles from LPS versus LPS + Dex treated macrophages for differential expression of read counts obtained at 1 kb around the 1684 eGBSs (Figure 1B). Actually, we found that GR ligand significantly induced 81 (adjusted *p*-value < 0.05, fold-change >1.5) and repressed 108 eRNAs (fold-change <−1.5) (Figure 3A).

The expression of these differential eRNAs was highly correlated with the expression of the closest gene (Spearman correlation coefficient = 0.81, Appendix A). Overrepresentation analysis for biological processes (Gene Ontology) indicated that most of the genes associated with decreasing eRNA expression (e.g., *Ccr3*, *Ccl5*, *Il1a*, *Il1b*) were involved in inflammatory processes such as ‘leukocyte migration’, ‘positive regulation of cytokine production’ and ‘positive regulation of inflammatory response,’ reflecting GR’s anti-inflammatory properties (Figure 3B, Appendix A). In contrast, eRNAs induced by Dexamethasone less significantly clustered into biological processes, most likely due to a lack of pathway commonalities between classically activated GR target genes such as *Tsc22d3* (*Gilz*), *Dusp1*, *Saa3* or *Edn1*. Still, genes associated with GR-induced eRNAs were enriched for processes involved in ‘leukocyte adhesion to the vascular endothelium,’ ‘regulation of blood pressure’ or ‘response to chemokines’ (Figure 3B, Appendix A).

Example genome browser tracks for the most highly induced and repressed GR eRNAs are displayed in Figure 3C. Interestingly, GR-mediated induction or repression of eRNAs was not limited to the GR binding site (GR ChIP-seq in Figure 3C), but extends to a wider region. These observations were confirmed by qRT-PCR for these eRNAs (Figure 3D), which may favor a model of GR-induced or -repressed local chromatin de-compaction [82]. Thereby, GR might allow intergenic access for RNA polymerase II and enable subsequent unstable transcription.

GR’s ability to simultaneously activate and repress gene expression in the same cellular context appeared to be reflected on the eRNA level. Hence, we asked whether we could find discriminatory features within the DNA sequences of those GR binding sites that were inducing or repressing eRNA transcription. We performed differential motif enrichment analysis using XSTREME [68], either searching for motifs that were significantly enriched within 100 bp of GBSs inducing over those repressing eRNA expression, or vice-versa. We could only find a significant enrichment (E < 0.001) for the GRE motif among the sites activating eRNA expression (Figure 3E). These results are in concordance with previous reports that showed that the classical palindromic GRE is usually found in cis-regulatory regions associated with activated GR target genes [19,83]. We note here that this particular GRE motif, however, slightly diverted from the classical palindromic consensus (AGAACAnnnTGTTCT) sequence—an indication for DNA-driven determination of regulatory polarity [84,85].

Altogether, our 4sU labelling revealed that both eRNA activation and repression responded to glucocorticoids in concordance with nearby target mRNAs, and that GR can both up- and down-regulate intergenic transcription from its binding site.

### 3.4. Enhancer RNA Expression Correlates with BRD4 and H3K27ac at GR Enhancer in Macrophages

As we had identified a number of activated and repressed eRNAs made from GR binding sites in response to Dexamethasone, we aimed to functionally characterize the connection between GR occupancy and the regulation of eRNA transcription. Therefore, we analyzed publicly available data for various chromatin features in LPS- and in LPS + Dex-treated macrophages. We only included data sets with at least two replicates per condition (please note that those datasets were generated in different laboratories and hence may differ with respect to treatment times and ligand concentrations. Please refer to Appendix A). We investigated open chromatin profiles by ATAC-seq and ChIP-seq for histone 3 lysine 27 acetylation (H3K27ac), and histone 3 lysine 4 (H3K4) mono- (me1), di- (me2) and tri-methylation (me3). We also included ChIP-seq data for the histone methyl transferase SETD1A, its co-factor CXXC1, the bromodomain protein BRD4 and RNA polymerase II [16,19].

First, we quantified read counts within 1 kb of all eGBSs with differentially expressed eRNAs (81 induced, 107 repressed, Figure 3A) and computed the changes in chromatin features or co-regulator occupancy between LPS + Dex versus LPS-only treated macrophages using DESeq2 [41]. We used the reads-in-peaks ratio as size factors for normalization to library size and ChIP efficiency (Appendix A, see method section for details). Afterwards, we determined those chromatin features and co-regulators whose Dex-dependent regulation correlated with the change in eRNA expression at differentially expressed eGBSs. We observed a positive correlation between eRNA expression and DNA accessibility (ATAC-seq), H3K27 acetylation and RNAP2, BRD4, SETD1A and CXXC1 recruitment (Figure 4A). These marks are well known for their association with gene activation [23,86,87,88,89,90]. The strongest positive correlation was observed with open chromatin (ATAC-seq) (Spearman correlation coefficient 0.78), H3K27ac (Spearman correlation coefficient 0.74) and BRD4 binding (Spearman correlation coefficient 0.74). In contrast, none of the three H3K4 methylation states correlated strongly with eRNA expression (Figure 4A).

To further determine the generalizability of these correlations, we plotted the log2-fold change distribution for each chromatin feature stratified by the corresponding change in eRNA expression (Figure 4B and Appendix A). We included those eGBSs that did not show a significant change (Benjamini–Hochberg-adjusted *p*-value > 0.05) in eRNA expression upon Dex treatment as a control group (see Figure 3A). All groups showed a broad distribution of eRNA expression changes, which was also reflected in a wider distribution of changes in chromatin features (Figure 4B). A closer look showed that eGBSs with Dex-induced eRNA expression also displayed increased DNA accessibility, H3K27 acetylation levels and RNA polymerase II and BRD4 occupancy. SETD1A and CXXC1 recruitment upon Dex-treatment, however, was only detected at a fraction of those enhancers, indicating that the recruitment of the SETD1A histone methyl transferase complex may be locus-specific and not a general feature associated with Dex-induced eRNA expression (Figure 4B) [19]. In line with these results, eGBSs with reduced eRNA expression upon Dex treatment showed reduced DNA accessibility, H3K27 acetylation, RNA polymerase II and BRD4 recruitment. Again, only a fraction of those repressed eGBSs appeared to lose SETD1A or CXXC1 occupancy (Figure 4B). We noticed that the changes observed at repressed eGBSs were less pronounced (lower fold changes), and fewer eGBSs showed a significant reduction (*p* < 0.1) in chromatin features when compared to activated eGBSs (Figure 4B). The H3K4 methylation status of GR binding sites with either induced or reduced eRNA expression appeared to be unaltered upon Dex treatment (Appendix A), in line with the weak correlation seen in Figure 4A.

Heatmaps displaying the mean ChIP-seq signal intensity of each of the profiled chromatin features, co-regulators and eRNA expression confirmed that GR-bound enhancers with induced eRNA expression showed increased DNA accessibility (ATAC-seq), H3K27 acetylation, RNA polymerase II, BRD4, SETD1A and CXXC1 occupancy in the presence of GR ligand (Figure 4C). Among the repressed eGBSs, we could only observe a minor reduction in H3K27ac and BRD4 occupancy by this method (Figure 4C). These observations are in line with the mild effects on open chromatin (only one site with more than 1.63-fold reduction in ATAC-seq signal) and the weak RNA polymerase II signal at enhancers (Figure 4B,D). Please note that the top and bottom fractions of the unaffected eRNAs, sorted by descending log2-fold change in transcription, mirror those patterns (although non-significantly) (Figure 4B,C). Again, we did not observe any effects on H3K4 methylation dynamics (Appendix A).

Finally, we chose two genes each associated with induced (*Tsc22d3* (*Gilz*), *Dusp1*) and reduced (*Ccr3*, *Il1a*) eRNA expression from GR binding sites for a more detailed look (Figure 4D and Appendix A). We observed an increase in DNA accessibility, H3K27 acetylation, RNA polymerase II, BRD4, SETD1A and CXXC1 recruitment at loci with induced eRNA expression (Figure 4D and Appendix A). These findings are in agreement with BRD4′s ability to bind acetylated histones via its bromodomain, and its involvement in gene activation, *p*-TEFb recruitment and subsequent RNA polymerase II pause release [90,91].

At repressed eGBSs, on the other hand, we observed a reduction in H3K27 acetylation and a loss of BRD4 occupancy, whereas RNA polymerase II was barely detectable at the GR-bound sites (Figure 4D and Appendix A). We did not observe significant changes in SETD1A recruitment or DNA accessibility, and only a few enhancers displayed reduced CXXC1 occupancy, in concordance with previous studies [19,36].

In conclusion, we showed that open chromatin, H3K27 acetylation, BRD4 and RNA polymerase II recruitment appeared to be the main factors associated with the regulation of eRNA expression from GR binding sites in murine macrophages. For a subset of activated GR-bound loci, a gain in DNA accessibility as well as SETD1A and CXXC1 recruitment may play additional roles in eRNA transcription and corresponding target gene activation, potentially implicating these factors in locus-specific gene control. Overall, H3K27ac and BRD4 recruitment consistently correlated with eRNA expression at activated and at repressed sites, suggesting a putative role for histone acetylation as a driver of eRNA transcription, possibly by keeping the chromatin in an uncondensed ‘open’ state.

## 4. Discussion

For decades, glucocorticoids have been widely used for the treatment of inflammatory, autoimmune and respiratory disorders, as well as COVID-19 and cancer. Interestingly, the molecular mechanisms of glucocorticoid-mediated transcriptional regulation are diverse and present a complex molecular puzzle. In this study, we aimed to globally profile enhancer RNAs (eRNAs) produced in response to GC treatment to gain mechanistic insight into the regulatory events upon GR DNA binding.

We found that GR-bound cis-regulatory loci newly synthesize unstable and lowly expressed non-coding RNAs, termed enhancer RNAs (eRNAs), in murine macrophages and livers [92,93]. Our profiles of eRNA expression from GR binding sites agree with reports from BEAS-2B epithelial cells stimulated with TNFα and Dexamethasone, and from A549 and U2OS cancer cell lines [94,95]. We found that GR-driven eRNA transcription occurred under physiological (liver in response to diurnal changes in glucocorticoid release) and inflammatory (activated macrophages) conditions.

In macrophages, almost 70% of GR binding sites synthesized detectable amounts of eRNAs (Appendix A). The distinction between eRNA-producing and non-productive sites may discriminate between functional and non-functional GR binding sites [96]. Motif enrichment analyses, however, did not yield significant differences in DNA motif composition between loci with or without eRNA transcription, although a tendency towards an increased AP1 motif occurrence at productive sites was observed (Figure 1D).

Moreover, eRNAs are expressed in a tissue specific manner, thereby providing a link between cell type-specific GR binding and function [29,32,76,97]. Indeed, we could confirm that eRNA expression in response to GCs was cell-type-specific. Tissue-specific eRNA-producing GR binding sites (eGBSs) were enriched for pioneer factors associated with the corresponding lineage, concordant with the requirement of a pre-determined accessible chromatin landscape for GR binding [28,31,32,75]. Conceivably, eRNAs might present interesting future therapeutic targets. For instance, the reduction in specific eRNAs might prevent adverse effects of glucocorticoid treatment or increased glucocorticoid secretion, which could be attributed to the activation of certain metabolic genes in selected tissues. Furthermore, eRNAs may be potential therapeutic options for inflammatory diseases, especially since reports showed that knockdown of the eRNA regulating *Ccl2* prevented systemic inflammation in mice [98]. Similarly, knockdown of the *Il1b* eRNA dampened IL1β production in a human monocytic cell line stimulated with LPS [99].

Differential expression analyses in LPS + Dex- versus LPS-stimulated macrophages revealed sites of eRNA activation and repression from GR-bound enhancers, in line with GR’s potential to both up- and down-regulate its target genes [4,16,19]. The changes in eRNA transcription were highly correlated with differential expression of their closest genes [97,99]. Functionally, genes associated with eRNA repression were involved in inflammatory processes, agreeing with GR’s anti-inflammatory properties in macrophages [4,19]. Moreover, we found classical GRE consensus sequences associated with induced eRNA expression, while no specific motif was enriched among binding sites with reduced eRNA production (Figure 3E). A subset of GR binding sites with eRNA upregulation containing GREs also showed increased SETD1A and CXXC1 recruitment and DNA accessibility upon GC treatment [19,36]. These cis-regulatory sites may be established de novo by GR, whereas the majority of sites may be pre-established and opened preceding GR recruitment, as indicated by existing P300 occupancy and H3K27 acetylation [14,19,36,75]. Which co-factors or DNA sequences determine the regulatory polarity of GR-mediated transcription regarding both eRNAs and mRNAs remains to be investigated. A putative mechanism underlying eRNA and mRNA repression in the absence of a transcription factor motif was proposed for the estrogen receptor (ER), a close relative of GR. ER-mediated gene repression, as well as recruitment to repressive ER binding sites, was found to be dependent on eRNA expression [100].

We found eRNA transcription closely correlated with enhancer activation, H3K27 acetylation and BRD4 recruitment [75,89,101,102]. Those results confirm previous observations implying eRNA expression in the recruitment of the histone acetyl transferase CBP with subsequent H3K27 acetylation [103]. H3K27 acetylation, in turn, may serve as an interaction surface and recruit BRD4 to cis-regulatory elements [89]. At promoters, BRD4 is able to recruit P-TEFb and subsequently induce RNA polymerase II pause release and productive transcription [88,90,91]. BRD4 recruitment at the enhancer may therefore induce activation of the associated promoter via 3D genome interactions. Additionally, BRD4 may bind eRNAs directly via its bromodomains and, thereby, stabilize an active enhancer conformation to maintain gene activation [104]. Recently, BRD4 was implied in the formation of transcriptional condensates that mediate enhancer–promoter interactions, especially at transcription factor binding hubs [105]. Such GR binding hubs may indeed exist (Figure 4D and Appendix A) [14,75,96,105]. However, the exact mechanisms underlying BRD4-mediated promoter–enhancer interactions and transcription initiation are only beginning to be elucidated [14,106,107]. Inhibiting BRD4 recruitment to acetylated chromatin using I-BET showed that BRD4 was required for the activation of inflammatory genes such as *Il1a* and *Il1b* in murine macrophages, and that GC treatment reduces the BRD4 occupancy at those promoters [16]. A decrease in histone acetylation at GR binding sites involved in repression has been observed in macrophages [4,16,19,36,75]. Reduced eRNA synthesis and the subsequent release of histone acetyl transferases, or GR-mediated recruitment of histone deacetylases, may explain the loss of H3K27 acetylation upon GR binding at repressive enhancers [4,13,27,104]; a mechanism that was previously described for the ER [102]. eRNA expression at ER binding sites correlated with gene activation, and eRNA knockdown abolished the response to estrogens, conceivably suggesting a functional role for eRNAs in nuclear receptor-mediated gene regulation in general [108].

In addition to BRD4, eRNAs are known to scavenge NELF from paused RNA polymerase II, thus enabling RNA polymerase II phosphorylation by P-TEFb and the activation of pause-controlled genes [91]. The GR has been shown to regulate paused and non-paused genes in murine macrophages. Furthermore, the repression of pause-regulated genes by glucocorticoids was found to depend on NELF recruitment followed by the accumulation of paused RNA polymerase II at their promoters [16].

Taken together, we suggest that GR-mediated gene activation and repression involve tissue-specific changes in eRNA transcription at a subset of GR-bound sites, coincident with BRD4 occupancy and histone acetylation, together with locus-specific regulation [16,19,109]. Whether all of these eRNAs actually have functional roles in gene regulation remains to be seen. One of the main arguments against such a function lies in the low expression level and short life span of enhancer RNAs. However, in light of the re-emerging role of condensates or transcriptional hubs in gene regulation, we point out that the local concentration of eRNAs at the individual loci is unknown. Potentially, the unstable nature of enhancer RNAs might make them ideal transcriptional regulators in the context of quick and nimble cellular adaptations to environmental signals; for instance, inflammatory stimuli [110,111].

## Figures and Tables

**Figure 1 cells-11-00028-f001:**
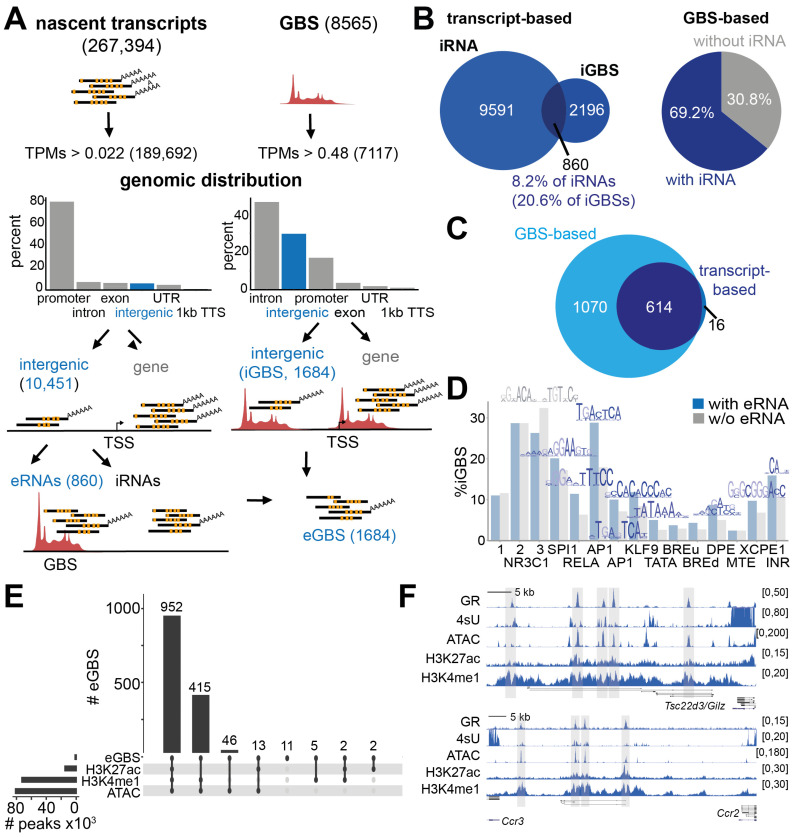
Nascent transcription from GR binding sites in activated macrophages. (**A**) Strategy for eRNA identification based on transcript expression (left, 4sU labeling) and GR binding sites (GBS) (right, ChIP-seq). Brackets show the number of reads or GBSs at each step. Bar plots display the overlap of nascent transcripts (left) or GR binding sites (GBS) with nascent transcription (right) with indicated genomic features. eRNA—enhancer RNA, eGBS—eRNA expressing GBS, iGBS—intergenic GBS, iRNA—intergenic RNA, TPMs—transcripts per million, TSS—transcriptional start site, TTS—transcriptional termination site, UTR—untranslated region. (**B**) Venn diagram indicating the overlap of iRNAs identified by the transcript-based approach with intergenic GBSs (left). The proportion of iGBSs with and without eRNA expression characterized by the GBS-based approach is shown on the right. (**C**) Venn diagram of the eGBS overlap between both strategies. One-thousand six-hundred and eighty-four RNAs identified by the GBS-centric method were used for further analyses. (**D**) Motif occurrence among intergenic GBSs (FIMO). Motifs enriched at iGBS with or without eRNA expression (see Appendix A) as well as motifs associated with promoters were included. (**E**) Overlap of eGBSs with classical enhancer features such as openness (ATAC-seq), H3K27ac and H3K4me1 (ChIP-seq). Horizontal bars: intersection size, vertical bars: peak set size. (**F**) Mean genome browser tracks for *Tsc22d3* (*Gilz*) and *Ccr2/Ccr3* showing nascent transcription (4sU, *n* = 3), GR binding (GR ChIP-seq, *n* = 2), open chromatin (ATAC-seq, *n* = 4), H3K27ac (ChIP-seq, *n* = 2) and H3K4me1 (ChIP-seq, *n* = 3). Data are from primary murine macrophages treated with LPS and Dexamethasone.

**Figure 2 cells-11-00028-f002:**
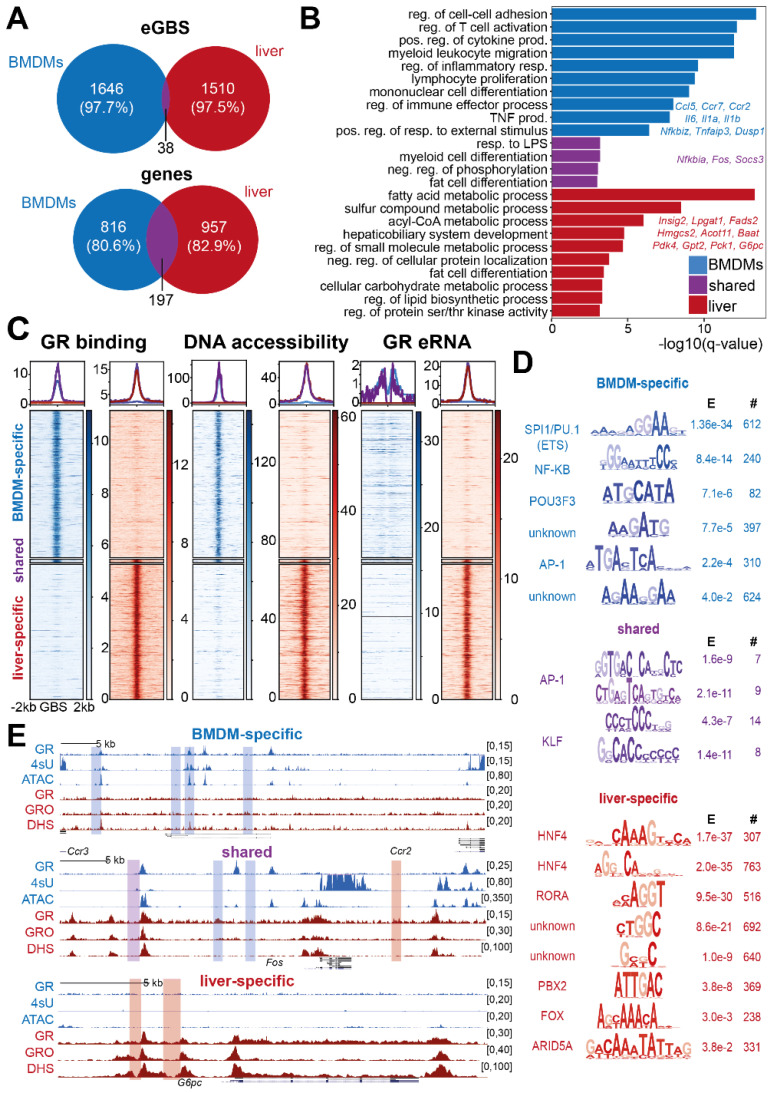
GR-mediated eRNA expression patterns are tissue specific. (**A**) Venn diagram for overlapping eRNA-expressing GR binding sites (eGBS) and their nearest target genes in liver and macrophages (BMDMs). (**B**) Functional annotation for the genes associated with eGBSs specific to macrophages or livers (top 10 non-redundant biological processes *q*-value < 0.001, with examples). neg. = negative, pos. = positive, prod. = production, reg. = regulation, resp. = response (**C**) Heatmap of GR ChIP-seq, open chromatin (macrophage ATAC-seq, liver DHS) and nascent transcription (macrophage 4sU, liver GRO-seq). Heatmaps display the mean signal intensity across biological replicates. Coverage plots show the median signal per subset (blue=macrophage-specific, red=liver-specific, purple=shared). (**D**) Differential XSTREME motif analysis [68] of macrophage-specific, shared and liver-specific eGBSs. E = E-value; # = number of sites. (**E**) Example genome browser tracks for macrophage-specific (*Ccr2/Ccr3*), shared (*Fos*) and liver-specific (*G6pc*) eGBSs showing mean GR ChIP-seq signal, nascent transcription profiled by 4sU-seq in macrophages and GRO-seq in livers and open chromatin (macrophage ATAC-seq, liver DHS) (GR ChIPseq *n* = 2, ATAC-seq *n* = 4, DHS *n* = 1, 4sU *n* = 3, GRO-seq *n* = 1).

**Figure 3 cells-11-00028-f003:**
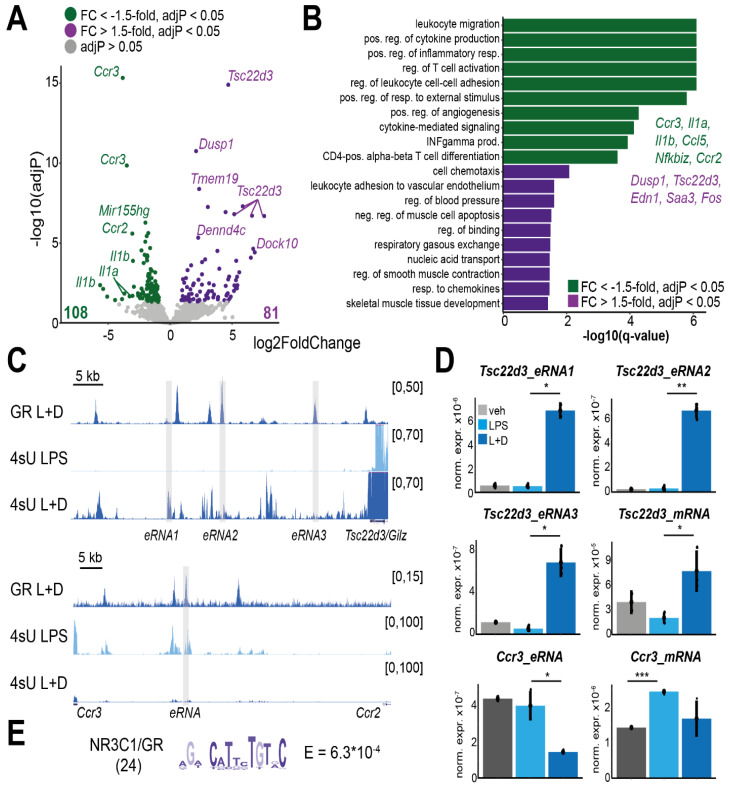
Glucocorticoids regulate differential eRNA expression. (**A**) Volcano plot displaying intergenic GBSs showing differential eRNA expression in LPS + Dex versus LPS-treated macrophages. Each dot represents one eRNA-producing GBS. Selected eRNAs are named by the closest TSS. (**B**) “Biological process” overrepresentation analysis of genes associated with Dex-induced or Dex-repressed eRNAs. Top 10 non-redundant processes with a q-value cutoff of 0.05 are shown and examples labelled. (**C**) Mean genome browser tracks of GR ChIP-seq (LPS + Dex; L + D, *n* = 2) and 4sU-seq coverage for LPS and LPS + Dex treated macrophages (*n* = 3). Shades indicate positions of qRT-PCR primers for D. (**D**) RT-qPCR for *Tsc22d3* (*Gilz*) and *Ccr3* and their eRNAs in macrophages treated with vehicle, LPS or LPS + Dex (*n* = 3). *** *p* < 0.001, ** *p* < 0.01, * *p* < 0.05 (Students *t*-test) (**E**) Differential motif enrichment analysis for GR-bound sites with activated eRNA expression over those with reduced expression. (24: number of sites with motif matches).

**Figure 4 cells-11-00028-f004:**
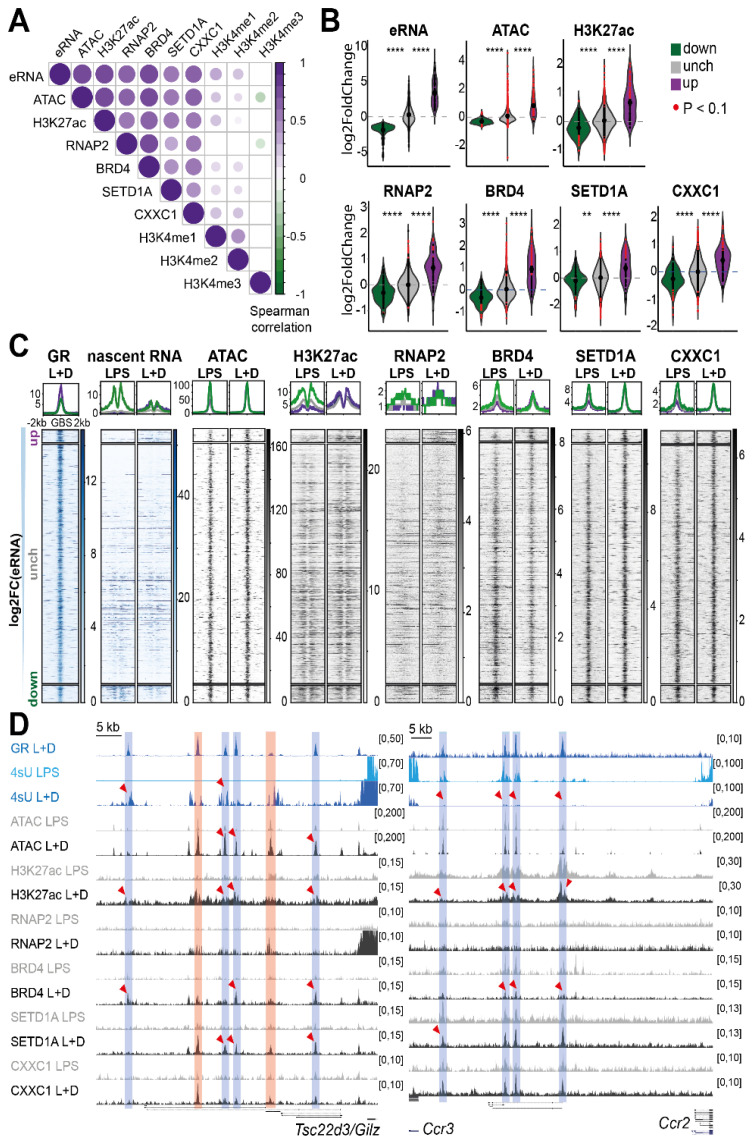
eRNA expression from GR binding sites correlates with H3K27ac and BRD4 recruitment. (**A**) Correlation of pairwise comparisons of histone marks and co-regulators between LPS and LPS + Dex stimulated macrophages at eGBSs. (**B**) Distribution of log2 fold-changes in chromatin features or co-regulators as Violin plot. eGBSs are grouped by significantly induced, repressed or non-significantly changed (adj. *p*-value > 0.05, Figure 3A). Each dot represents one enhancer. Red denotes significance (*p* < 0.1) (ChIP-seq, Wilcoxon-Mann-Whitney test, **** *p* < 0.00001, ** *p* < 0.001) (**C**) Mean heatmaps of ChIP-seq signals in LPS or LPS + Dex treated macrophages, grouped by eGBSs with induced, repressed or non-changing eRNA expression. eGBSs are sorted in descending order of the log2-fold change. Coverage plots above summarize median ChIP-seq densities. Colors as indicated in B. (**D**) Genome browser tracks of induced (*Tsc22d3*) and repressed (*Ccr3*) eRNAs, means from biological replicates. Red shades mark eGBSs with alterations of all displayed features. Red arrows point at significant signal changes at eGBSs (blue shades). (ATAC-seq *n* = 4; 4sU-seq *n* = 3; H3K27ac, RNAP2, BRD4, SETD1A, CXXC1 and GR *n* = 2 ChIP-seq).

## Data Availability

The 4sU-seq data is available at GEO (https://www.ncbi.nlm.nih.gov/geo/, accessed on 10 December 2021) with the accession number GSE186629. Further publically available data integrated in this study is deposited at GEO with the following accession numbers: GSM5654968-75; GSM4040428-54; GSM4040456/57; GSM4040459-62; GSM4040483-86; GSM2911260/61; GSM1479704; GSM1437739; GSM2911206-08; GSM2911215-17 and GSM4078778-83. For more details refer to Appendix A. R scripts are deposited on github (https://github.com/FranziG/GReRNAs, accessed on 10 December 2021) and are provided upon request. All software used in this manuscript was last pulled from their respected web source in July 2021.

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
