# Peer review of "Enhancer RNA Expression in Response to Glucocorticoid Treatment in Murine Macrophages"

_cells, 2021, doi:10.3390/cells11010028_

Round 1

Reviewer 1 Report

The authors identified and characterized the enhancer RNAs after glucocorticoid stimulation in LPS-activated primary murine macrophages using 4sU-seq. The GR-mediated eRNAs in H3K27 acetylation and BRD4 recruitment were also determined. This manuscript was well designed and the data can fully support the results. I suggest this manuscript could be accepted for publication after minor revisions.

Minor issues

  1. The abstract was not well written. No any results were provided, please summarize the data and results and cooperated into the second paragraph of abstract.
  2. Line 104: 5% CO2 should be subscripted.
  3. Line 141-147: different font size was used.
  4. Line 154-163: please provide more detail information about the primers, such as product length, Tm, source etc.
  5. Line 660 and Line 663: Same abbreviation, eRNAs, was used for enhancer RNAs and expressed non-coding RNAs. Please carefully revise and point out what eRNAs meaning for after Line 663.
  6. Lack of conclusions at the end of disscussion.

Author Response

We thank the reviewer for his/her comments.

Concerning the abstract: The second paragraph of the abstract contains the following part stating our results:

“Here, we have profiled nascent transcription upon glucocorticoid stimulation in LPS-activated primary murine macrophages using 4sU-seq. We compared our results to publicly available nascent transcriptomics data from murine liver and bioinformatically identified non-coding RNAs transcribed from intergenic GR binding sites in a tissue-specific fashion. These tissue-specific enhancer RNAs (eRNAs) correlate with target gene expression, reflecting cell type-specific glucocorticoid responses. We further associate GR-mediated eRNA expression with changes in H3K27 acetylation and BRD4 recruitment in inflammatory macrophages upon glucocorticoid treatment.”

Additionally, we have made the following changes to the text as suggested by the reviewer:

Line 104: We changed “5% CO2 “ to “5% CO2”.

Line 141-147. We have adjusted the font size.

Line 154-163. We have added information on annealing temperatures and PCR product length for all primer pairs (see below).

Tsc22d3/Gilz (Tsc22d3_F: ACCACCTGATGTACGCTGTG; Tsc22d3_R: TCTGCTCCTTTAGGACCTCCA, TA=60.0°C, 51 bp product), Tsc22d3/Gilz eRNAs 1 (Tsc22d3_eRNA1_F: TTGAGCAGCAGACACGTTCA; Tsc22d3_eRNA1_R: GAATTCCTCCGACGGGAACT, TA=60.2°C, 189 bp product), 2 (Tsc22d3_eRNA2_F: AACACAGGGTGGACATTGGG ; Tsc22d3_eRNA2_R: ATGGTGGTGGAAAGCAAGGT, TA=60.2°C, 73 bp product) and 3 (Tsc22d3_eRNA3_F: GGAGTTGGG TTCTGCCTGAG, Tsc22d3_eRNA3_R: CCTGACCCTGTCTTTCCAGTG, TA=60.3°C, 52 bp product), Ccr3 (Ccr3_F: GAAAACTTGCAAAACCTGAGAAGC; Ccr3_R: TGCCATTCTACTTGTCTCTGGTG, TA=59.7°C, 174 bp product), Ccr3 eRNA (Ccr3_eRNA_F: TGTAGAGCAGAGGGCTGGAT, Ccr3_eRNA_R: CCAGGTTACAGTGCCCCATT, TA=60.0°C, 98 bp product) and the housekeeping gene 18S (18S_F: AAACGGCTACCACATCCAAG, 18S_R: CCTCCAATGGATCCTCGTTA, TA=58.2°C, 155 bp product)”

We also specified that all primers are targeting the murine genome in line 152 (“murine genes or genomic loci”). We also added the primer provider and target range for the annealing temperatures in line 165/166 (“Primers were designed to have annealing temperatures (TA) between 58 and 61°C  (Eurofins).”).

Line 660 and 663: We have changed the paragraph as follows to clarify that we call unstable, lowly expressed non-coding RNAs from GR binding sites eRNAs. The new sentence states this as follows:

“We found that GR-bound cis-regulatory loci newly synthesize unstable and lowly expressed non-coding RNAs, termed enhancer RNAs (eRNAs) in murine macrophages and livers”

At the end of the discussion, we concluded our study with the statement “Taken together, we suggest that GR-mediated gene activation and repression involve tissue-specific changes in eRNA transcription at a subset of GR-bound sites, coincident with BRD4 occupancy and histone acetylation, together with locus-specific regulation.” Followed by a small section with open questions for further scientific discourse.

Reviewer 2 Report

Overall:  The manuscript is well written, and the results are interesting. I have no real concerns with the experimental design, although the paper could have been stronger with the addition of some more mechanistic studies.  That said, the data shown is state of the art and will have considerable impact.

Minor points:  While I appreciate the detailed materials and methods section, I think it could be edited for length. There are also places where the font size is different and minor typos that need addressed.  There is also too much of the data buried in the supplement, which should be moved into the main text, especially Figures S1 and S2, which are central for the story. 

Author Response

We thank the reviewer for her/his comments. We have corrected the different font sizes. Concerning the length of the method section, we prefer the detailed description for reproducibility purposes. Nevertheless, we removed the following parts to shorten the methods section:

Lines 111-117: “2 ml of chloroform were added to 10 ml of cell lysate, mixed vigorously and incubated for 2-3 min at 20°C. After centrifugation (15 min, 13,000xg at 4°C), the upper aqueous phase was transferred to a new 15-ml polypropylene tube. The RNA was precipitated with 0.5 volumes of RNA precipitation buffer (1.2 M NaCl, 0.8 M sodium citrate) and 0.5 volumes of isopropanol and washed twice with 75% ethanol (centrifugation for 10 min at 13,000xg and 4°C). The RNA was resolved in RNase-free water and the concentration was measured using a Nanodrop1000 (ThermoFisher Scientific).”

Lines 125-130: “Shortly, columns were pre-equilibrated in washing buffer (100 mM Tris pH=7.4, 10 mM EDTA, 1 M NaCl, 0.1 % Tween-20), 100 ml streptavidin beads were added to 100 ml 1 mg/ml biotinylated RNA and loaded onto the column. Beads were washed three times in pre-warmed (65°C) washing buffer and three times in washing buffer at 20°C. RNA was eluted twice with 100 ml of 100 mM DTT into 400 ml of magnetic Agencourt RNA clean XP beads.”

Lines 131-135: “as follows: 5 min incubation at 20 °C, 5 min incubation on the magnetic rack at 20 °C, removal of the supernatant, 1x wash with 1 ml 70% ethanol, 5 min incubation on the magnetic rack at 20 °C, removal of ethanol, air-drying Agencourt beads for approximately 10 min at 20 °C, elution in 30 ml RNase-free water.”

Lines 169-188: We replaced paragraph 2.6.1 with the shorter version below by referring to table S5, which now also includes the GEO accession numbers for all samples.

“ChIP-seq against GR, H3K4me1/me2/me3, H3K27ac, SETD1A, CXXC1, ATAC-seq and RNA-seq in bone marrow-derived macrophages was previously published by our lab and is available at the gene expression omnibus (GEO). Similarly, our lab previously released ChIP-seq data against GR (GSM2911260 and GSM2911261 [3]) and RNA-seq data from murine liver tissue (GSE108690 [3]).

We further used publicly available ChIP-seq data for RNAP2 and BRD4 in murine bone marrow-derived macrophages (GSE109131) and DNaseI HyperSensitivity (DHS) (GSM1479704) as well as GRO-seq data for murine livers (GSM1437739). An overview of the data including individual accession numbers and exact treatment conditions appears in table S5.” 

As suggested by the reviewer, we corrected several typos:

Line 104: We change “5% CO2 “ to “5% CO2”.

Line 151: “were” - “was”

Line 181: “DnaseI” - “DNaseI”

Line 201: “sU-seq” - “4sU-seq”

Concerning the supplementary figure 1, we have included the previous figures S1A (into new figure 1A), S1C and S1F (into new Figure 1B) into the main figure 1 and changed the figure legends and text accordingly.

As for Figure S2, a comparison of gene expression, GR binding sites and DNA accessibility in macrophages and livers after glucocorticoid stimulation (shown in Figure S2) was previously published and discussed (Greulich and Hemmer 2016). Therefore, we do not believe that this information will provide novel insights to the reader and rather represent a proof of principle. With respect to the already very crowded Figure 2, we decided to keep the content of figure S2 in the supplement.

Reviewer 3 Report

Review report:

“Enhancer RNA expression in response to glucocorticoid treatment in murine macrophages”

In this work the authors analyzed the transcription of glucocorticoid-induced genes stimulated with LPS in murine macrophages.

This is a very accurate study, since the RNA-seq is very susceptible to the viability of the cells, I would like to ask the authors if, after having isolated of bone-marrow-derived macrophages, they evaluated their viability, and if so with what method.

Author Response

We thank the reviewer for the question. We tested the viability of our cells by visual inspection while establishing the 4sU assay (testing for the optimal labelling concentration). We used the 4sU concentration with an optimal labelling efficiency (tested by dot plot) and no visible impact on cell viability. Furthermore, we ensured the use of high quality (non-degraded) RNA as determined by Bioanalyzer for all our experiments (RIN 7.7-8.6). Additionally, we did not observe apoptosis or cell death as pathways among the differentially expressed genes.

Reviewer 4 Report

In the current manuscript Greulich et al. study the tissue specific regulation of glucocorticoid receptor downstream sigalling and  targeting and identified non-coding RNAs transcribed from intergenic  GR binding sites in a tissue-specific fashion. They further associate GR-mediated eRNA expression with changes in H3K27 acetylation and BRD4 recruitment in inflammatory macrophages upon glucocorticoid treatment.

In conclusion they propose a common mechanism by which GR-bound enhancers regulate target gene expression by changes in histone acetylation, BRD4 recruitment and eRNA expression and argue that local eRNAs are potential therapeutic targets downstream of GR signaling which may modulate glucocorticoid response in a cell type-specific way.

The manuscript reads well, contains a comprehensive data set that fully supports the conclusions regarding this innovative mechanism of tissue specific effects of glucocorticoids. I have no detailed comments on this comprehensive manuscript.

Author Response

We sincerely thank the reviewer for her/his time and effort, and the nice words. : )